# Heterogeneity-Based Management Restores Diversity and Alters Vegetation Structure without Decreasing Invasive Grasses in Working Mixed-Grass Prairie

Cameron Duquette [1,*], Devan Allen McGranahan [2], Megan Wanchuk [3], Torre Hovick [3], Ryan Limb [3] and Kevin Sedivec [3]

[1] Plant and Environmental Sciences, New Mexico State University, Las Cruces, NM 88003, USA
[2] USDA-ARS, Livestock & Range Research Laboratory, Miles City, MT 59301, USA; devan.mcgranahan@usda.gov
[3] Range Science, North Dakota State University, Fargo, ND 58105, USA; megan.wanchuk@ndsu.edu (M.W.); torre.hovick@ndsu.edu (T.H.); ryan.limb@ndsu.edu (R.L.); kevin.sedivec@ndsu.edu (K.S.)
* Correspondence: cduquett@nmsu.edu

**Abstract:** Non-native plants can reduce grassland biodiversity, degrade wildlife habitat, and threaten rural livelihoods. Management can be costly, and the successful eradication of undesirable species does not guarantee the restoration of ecosystem service delivery. An alternative to the eradication of invasive species in rangelands is to target the restoration of diversity and heterogeneous plant structure, which have direct links to ecosystem function. In this study, we evaluate patch-burn grazing (PBG) with one and two fires per year and variably stocked rotational grazing in *Poa pratensis*- and *Bromus inermis*-invaded grasslands using traditional (cover) and process-based (diversity and vegetation structural heterogeneity) frameworks in central North Dakota, USA. Within 3–4 years of initiating management, we found little evidence of decreased *Poa pratensis* and *Bromus inermis* cover compared to continuous grazing (*Poa pratensis* $F_{3,12} = 0.662$, $p = 0.59$; *Bromus inermis* $F_{3,12} = 0.13$, $p = 0.13$). However, beta diversity increased over time in all treatments compared to continuous grazing ($t_{PBG1} = 2.71$, $t_{PBG2} = 3.45$, $t_{Rotational} = 3.72$), and variably stocked rotational treatments had greater increases in spatial heterogeneity in litter depth and vegetation structure than continuously grazed pastures ($t_{visual\ obstruction} = 2.42$, $p = 0.03$; $t_{litter\ depth} = 2.59$, $p = 0.02$) over the same time period. Alternative frameworks that promote grassland diversity and heterogeneity support the restoration of ecological services and processes in invaded grasslands.

**Keywords:** patch-burn grazing; heterogeneity; grassland ecosystem processes

## 1. Introduction

Grasslands are important repositories of biodiversity and sources of income, providing myriad ecosystem functions and services. However, the extent of grasslands worldwide has declined substantially due to land use change, placing a large burden on those that persist. Remaining grasslands are impacted heavily by non-native plant invasion, which alters the plant–pollinator networks, reduces vegetation diversity and structural complexity, and drives ecological state conversion [1–6]. As the problem of grassland invasion has gained visibility, landowners and managers have realized the urgency and importance of managing to control invasive species' impacts.

Invasive species control measures are numerous and have achieved varying degrees of success. For example, while the targeted application of herbicides can be effective, many invasive plants have similar physiological characteristics to native species, increasing the risk of off-target mortality [7]. Mechanical control is labor intensive and often infeasible at broad scales due to the cost and the necessity of frequent repeat treatments [8]. The use of exotic natural enemies to control invasive plants has achieved success in some cases,

but full control is rare, and natural enemies have the potential to become invasive species themselves [9]. Though varied in scope, cost, and mechanism, these techniques inherently rely on abundance measures of the invasive species in question to determine success.

Using invasive species abundance and percent coverage as the measures of success in controlling non-native species is intuitively appealing, but these metrics may not closely align with higher-level management goals. On one hand, they are straightforward and objective, and are correlated with properties such as propagule pressure and invasive regime resilience [10,11]. In addition, the logical link between techniques such as targeted herbicide application or mechanical removal and the abundance or cover of undesirable species is clear. But on the other hand, percent cover is essentially a proxy measurement, whereas the true objective of process-based management is to mitigate the undesirable effects of invasive species on rangeland systems. Direct invasive species control can actually diminish grassland function if native plants do not recover to replace them [12]; in fact, many introduced species initially restored function to the degraded ecosystems they were brought into [13]. In these cases, it is possible to successfully eradicate invasive species while also having a neutral or negative impact on important ecological characteristics, such as biodiversity and community stability [14,15]. Instead of controlling invasive species directly by reducing their coverage or abundance, some managers have pursued alternative goals, such as targeting the restoration of invaded grassland function through the reinstatement of ecological processes.

A process-based approach to rangeland conservation centers on restoring spatial heterogeneity and biodiversity by mimicking pre-colonial disturbance processes [16]. Specifically, the Great Plains grasslands were structurally diverse ecosystems: spatially-patchy fires attracted native ungulates (primarily bison, *Bison bison*) that focused their grazing on high-quality forage in new burns, maintaining short-stature vegetation with ample bare ground [17]. Focused grazing pressure on recently-burned areas allows fuel to accumulate in unburned portions of the landscape, increasing the likelihood of subsequent fires. Through time, this creates a shifting mosaic of vegetation structure that increases niche diversity and maintains intact grassland systems [18,19]. Extensive research has shown that these benefits can be realized using prescribed fire and cattle grazing (patch-burn grazing, PBG) as proxies for the interaction between precolonial fire and native ungulate grazing (pyric herbivory) [17,20–22]. Furthermore, restoring this native disturbance regime has shown promise in invasive species management. For example, patch-burn grazing reduced cover and restored ecological function by promoting structural heterogeneity in sericea lespedeza- (*Lespedeza cuneata*) invaded tallgrass prairies [23]. However, the degree to which pyric herbivory has the potential to meet each definition of control success (reducing/eliminating invasive cover versus restoring ecological processes) in C3-dominated grasslands remains unknown.

*Poa pratensis* and *Bromus inermis* are two invasive C3 grass species that have caused large changes to the ecosystem structure and function in the Northern Great Plains. Initially brought as a forage species in the 18th and 19th centuries [3], each has proliferated due to wet climate cycles, strong competitive ability, the promotion of 'rest is best' grazing practices, and the removal of historic disturbance regimes [24–26]. Today, 10% of the Great Plains grasslands are dominated by these two species [27], with coverage in the Northern Great Plains states reaching as high as 86% [3]. These invasions are associated with impacts such as lower grassland pollinator diversity and floral resource availability [28], limited livestock forage availability during times of drought [29], and altered microbial and arthropod communities [26]. Efforts to reduce invasive grass cover in the Northern Great Plains have been marginally successful, with a consensus among rangeland managers that C3 non-native species are likely here to stay [24]. However, by focusing limited available resources on the restoration of ecosystem function and process in the context of plant invasions, we may still be able to achieve the desired outcomes [23].

In this study, we evaluated the effects of restoring ecological disturbance on the abundance of undesirable species using process-focused grazing. Specifically, we investigated

whether pyric herbivory (implemented via patch-burn grazing) and modified rotational grazing were effective at reducing the cover of *Poa pratensis* and *Bromus inermis*, enhancing biodiversity, and restoring structural heterogeneity compared to conventional livestock management practices (continuous grazing). As the use of fire in range management does not have broad support in the Northern Great Plains [30], we sought to determine if we could mimic the heterogeneity-creating effects of fire and grazing without the use of fire via a modified rotational grazing system with variably stocked paddocks. We hypothesized that while pyric herbivory might not immediately reduce the cover of *Poa pratensis* and *Bromus inermis*, it would increase the contrast in vegetation structure among patches, which is an important proxy for biodiversity [31]. We hypothesized that modified rotational grazing would generate more heterogeneity in vegetation structure and biodiversity than continuous grazing, but less than patch-burn grazing. Our results inform future management efforts, cultivating focused and realistic objectives by which to measure the success of invasive species control.

## 2. Materials and Methods

### 2.1. Study Site

We conducted our study from 2017 to 2020 at the Central Grasslands Research Extension Center in the prairie pothole region of North Dakota, USA (99°25′ W, 42°42′ N). The region was historically dominated by native C3 grasses, such as *Nassella viridula* (Trin. Barkworth), *Heterostipa comata* (Trin. & Rupr.), and *Pascopyrum smithii* (Rydb.), but now *Poa pratensis* and *Bromus inermis* comprise the majority of vegetation cover [32,33]. Common forbs include *Solidago* spp., *Asclepias* spp., and *Cirsium* spp., while *Symphoricarpos occidentalis* (Hook.) is the dominant woody species [33]. Common non-native forb species include *Artemisia absinthium* (L.), *Melilotus officinalis* (L.), and *Medicago sativa* (L.). During the growing season (1 May to 1 September), daytime average temperatures are 17.32 °C and precipitation totals average 28.6 cm [33]. Growing season average daytime temperatures and precipitation during the study period were: 2017: 17.3 C, 22.6 cm; 2018: 18.6 C, 38.7 cm; 2019: 16.3 C, 33.4 cm; 2020: 18.1 C, 20.4 cm [34]. Dominant ecological sites included loamy, sandy, shallow gravel, thin loamy, and very shallow [34].

Our treatment structure consisted of four replicates (65 ha each) of four grazing treatments. Two treatments employed patch-burn grazing, and were designed to mimic the historic disturbances of wildfire and bison grazing. The first was burned once a year (1/4 pasture, 16.2 ha) in the dormant season (late April–early May; hereafter PBG1). The second was burned twice a year (1/8 pasture, 8.1 ha, 16.2 ha/year total; hereafter PBG2) in the dormant season and the late growing season (late August–late September). For each, fire return intervals were set at four years to mimic pre-colonial fire regimes.

We designed a modified rotational grazing treatment to determine whether grazing management could restore ecological function without the use of fire. Each modified rotational grazing pasture was divided into four paddocks with interior fencing. Each paddock was stocked for varying lengths of time each year, with the goal of generating heterogeneity in vegetation structure and reducing grazer selectivity. The stocking durations in the modified rotational paddocks for each year were 0 days (rested, 0% vegetation removal), 21 days (moderate, 20–40% vegetation removal), 50 days (full, 40–60% vegetation removal), and 77 days (heavy, 60–80% vegetation removal), divided between two intervals each year with 40% of the days grazed during the first rotation and 60% grazed during the second rotation. Grazing intensities rotated between paddocks each year (e.g., heavily-stocked paddocks transitioned to rest the following year) so that each paddock would receive each grazing intensity over a four-year period. Finally, the continuous grazing treatment represented traditional grazing practices with no fire or interior fencing.

Stocking rates in all treatments were between 2.26 and 2.31 animal unit months per hectare to achieve forage use rates of 40–50% removal of standing crop. The first year of data collection marked the beginning of patch-burn implementation, so we were able to monitor changes over time as pastures transitioned from unburned to a full cycle. Due

to weather constraints, two of the PBG2 dormant season burns were not conducted in 2018 and all four were not conducted in 2019. The modified rotational treatment was first implemented in 2018, so results only represent three years of this treatment. Patch-burn treatments were randomized, but continuous grazing treatments were sited due to management restrictions (i.e., this area was in an easement that restricted management with fire and additional fencing) and the rotational treatment was added retroactively. Vegetation sampling transects were distributed across all major ecological sites in each pasture. Refer to Appendix A for further explanation of treatment layout.

### 2.2. Data Collection

We conducted vegetation surveys each year during the peak of the growing season (early- to mid-July). For each pasture replicate, we established two parallel 150 m transects in each patch (1/8 pasture, 8.1 ha). Transect pairs were 30 m from each other. We established 10 vegetation monitoring points along each transect (every 15 m, 20 points total per 1/8 pasture). At each sample point, we measured visual obstruction using a Robel pole, using an average of 4 readings taken from a height of 1 m at a distance of 4 m in each cardinal direction [35]. Robel readings incorporate information on vegetation height and density into a measurement of vegetation structure [36]. We then deployed a 0.5 m² frame at each plot and measured litter depth in each corner using a ruler. Finally, we estimated the cover of vegetation groupings using the cover class midpoints of the Daubenmire cover class scale ([37]; 0%, 3%, 15%, 38%, 63%, 85%, 98%). We classified vegetation into the following groups: *Bromus inermis*, *Poa pratensis*, native C3 grasses, non-native C3 grasses (excluding *Bromus inermis* and *Poa pratensis*), native C4 grasses, non-native C4 grasses, native legumes, non-native legumes, native forbs, non-native forbs, native woody plants, non-native woody plants, bare ground, and standing dead vegetation. All observers were trained in methodology and plant identification each year, and observers calibrated cover estimations to each other to ensure consistent results.

### 2.3. Data Analysis

To assess the success of grazing management in controlling non-native C3 grasses, we first focused on evaluation using traditional metrics (i.e., changes in the cover of undesirable vegetation groups and/or cover of desirable vegetation groups). We made univariate comparisons of functional groups at the start and end of the study (change relative to continuous grazing) using linear mixed models with pasture as a random effect (R Core Team 2022). We then performed a nonmetric multidimensional scaling analysis on vegetation composition to determine vegetation community trajectories of pastures over time using the *vegan* package with Bray–Curtis distance measures [38,39]. We plotted vectors in the ordination space corresponding to the movement of pasture centroids from the beginning to the end of the study.

We then defined beta diversity as the space occupied by pastures in the ordination space (i.e., multivariate dispersion [40]). We calculated the distance to pasture centroids and used this metric for beta diversity calculations, as this measure represents the average variability in functional group composition among patches (1/8 pasture) for each pasture [40]. Differences in beta diversity among treatments were then evaluated using a homogeneity of dispersion test ('betadisper' in *vegan*) [39,40]. We then focused our analysis on attributes that are indicative of ecosystem functioning and service delivery, specifically structural attributes (litter cover, litter depth, bare ground, and visual obstruction), beta diversity, and spatial heterogeneity. First, changes in structural attributes over time relative to continuous grazing were carried out using linear models, similarly to the vegetation group comparisons. We defined spatial heterogeneity of vegetation structure as the degree of difference among patches for bare ground cover, litter cover, litter depth, and visual obstruction. We accomplished this using a random effects-only modeling approach using patch ID as a random effect with structural attributes as the response variable [41]. This method obtains the amount of variance attributable to the patch term (1/8 pasture) for

each treatment. We chose this metric to quantify spatial heterogeneity as opposed to the coefficient of variation (CV) because CV scales with the mean of the response, whereas we were interested in the absolute variance structure (i.e., small absolute differences in mean structure among recently-burned patches receive high importance in CV calculations) [42].

## 3. Results

*Bromus inermis* was 2–3 times more abundant on continuous grazing pastures compared to the other treatments, indicating likely pre-treatment site differences (Figure 1). Differences over the study period did not vary between treatments ($F_{3,12}$ = 0.13, $p$ = 0.13; Figure 2). *Poa pratensis* cover was similar across all treatments at the beginning of the study, and did not change over time ($F_{3,12}$ = 0.662, $p$ = 0.59; Figures 1 and 2).

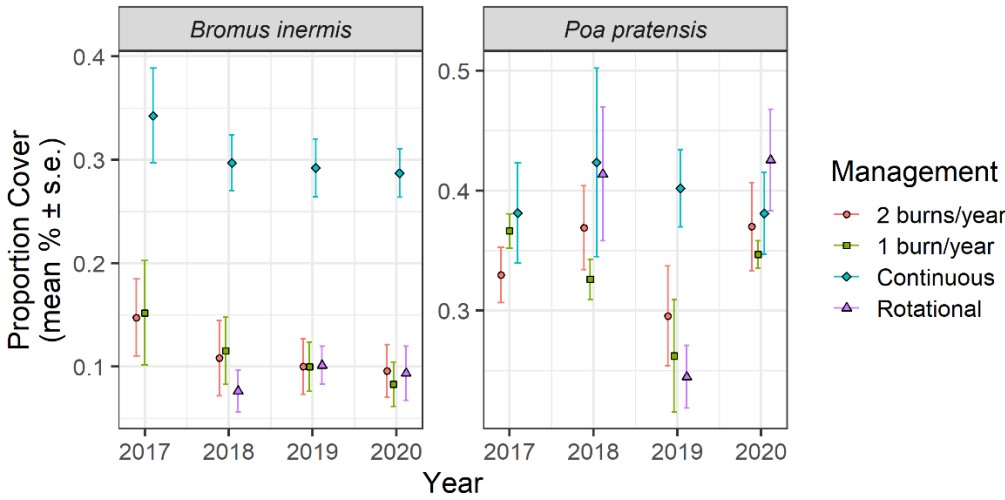

**Figure 1.** Cover of *Bromus inermis* and *Poa pratensis* over time in patch-burn treatments with one burn/year, two burns/year, continuous grazing, and modified rotational grazing systems from 2017 to 2020 at the Central Grasslands Research Extension Center in Streeter, ND. Error bars represent one standard error.

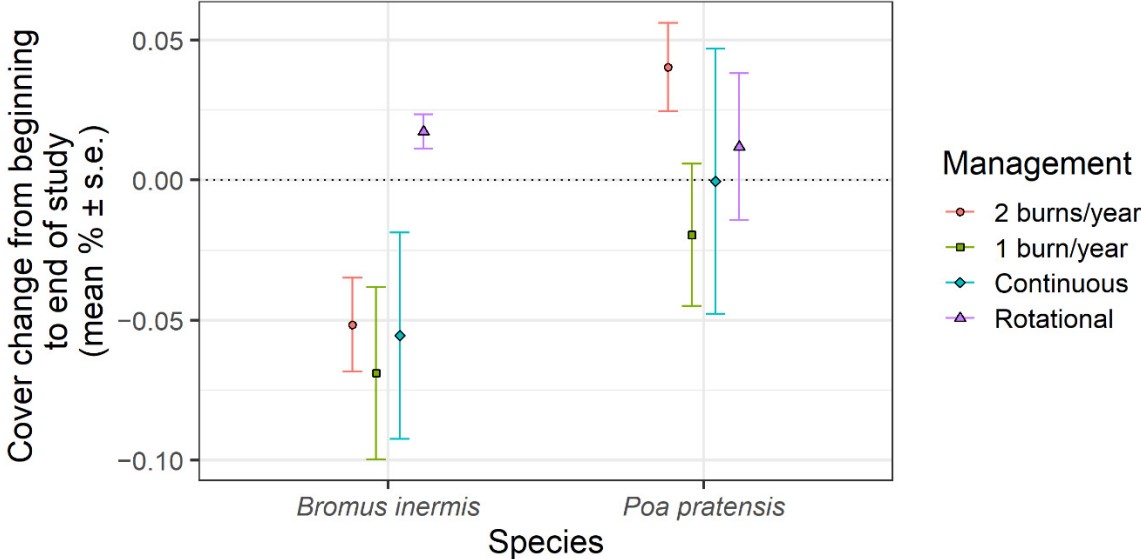

**Figure 2.** Changes to the cover of *Bromus inermis* and *Poa pratensis* in patch-burn treatments with one burn/year, two burns/year, continuous grazing, and modified rotational grazing systems from the beginning to the end of treatment implementation at the Central Grasslands Research Extension Center in Streeter, ND. Error bars represent one standard error.

Native C3 grass cover did not change over the study period ($F_{3,12}$ = 1.80, $p$ = 0.20; Figure 3).

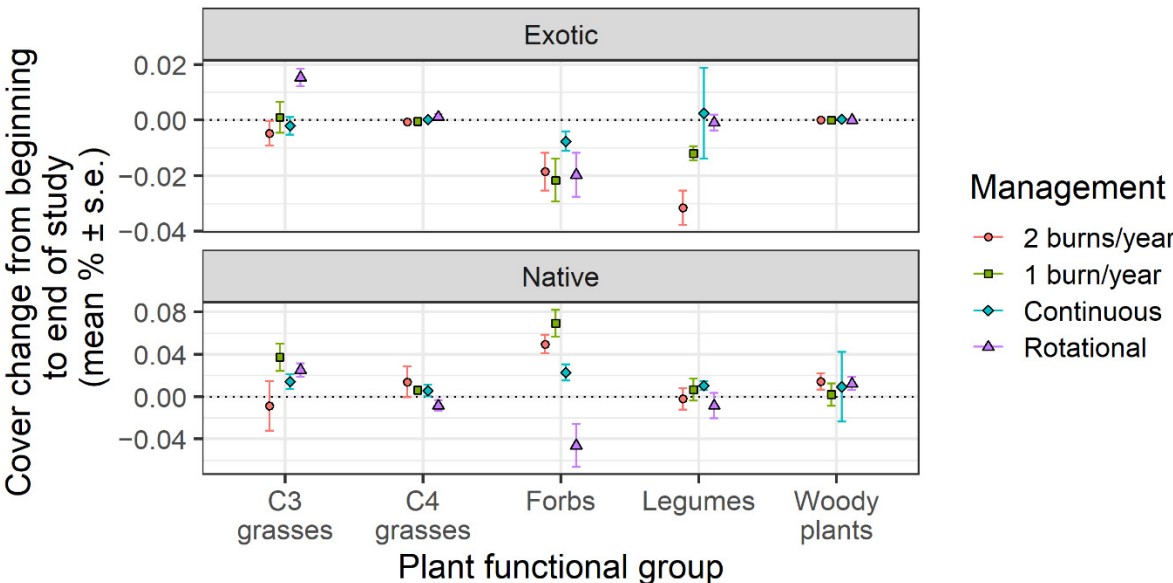

**Figure 3.** Changes to the cover of vegetation functional groups of native and exotic species in patch-burn treatments with one burn/year, two burns/year, continuous grazing, and modified rotational grazing systems from the beginning to the end of treatment implementation at the Central Grasslands Research Extension Center in Streeter, ND. Error bars represent one standard error.

Native forb cover declined over time in modified rotational grazing treatments relative to continuous pastures, while it increased in the PBG1 treatment ($t_{Rotational}$ = −3.66, $p$ < 0.01; $t_{PBG1}$ = 2.47, $p$ = 0.03; Figure 3). Native legumes and C4 grasses did not exhibit ecologically significant differences in cover over the study period ($F_{legumes\ 3,12}$ = 0.793, $p$ = 0.52; $F_{C4\ 3,12}$ = 1233, $p$ = 0.34; Figure 3). Native woody cover did not change relative to continuous grazing pastures over the study period ($F_{3,}12$ = 0.09, $p$ = 0.96; Figure 3). Non-native C3 (e.g., *Elymus repens* L. *Agropyron cristatum* L. Gaertn.) did not change significantly relative to continuous pastures, though we saw marginal increases in modified rotational pastures ($F_{3,12}$= 2.31, $p$ = 0.13; $t_{Rotational}$ = 2.09, $p$ = 0.06; Figure 3). Exotic forbs (e.g., *Cirsium arvense* L., *Artemisia absinthium* L.) decreased in all treatments, with no differences relative to continuous pastures ($F_{3,12}$ = 0.88, $p$ = 0.48; Figure 3). Exotic legumes (e.g., *Melilotus officinalis* and *Medicago sativa*), decreased in patch-burn treatments relative to control pastures ($t$ = −2.69, $p$ = 0.01; Figure 3). Exotic C4 grasses and woody plants constituted a minority of the vegetation cover across all the treatments, so we were not able to assess their changes.

Overall, the composition of patch-burn pastures changed towards more cover by native species following the treatment application (Figure 4). The composition of modified rotational pastures was largely stable over time, while the composition of continuous grazing pastures was not consistent (Figure 4).

Beta diversity was initially higher in the continuously grazed pastures compared to both PBG and modified rotationally grazed pastures (Figure 5). However, beta diversity in the continuous pastures declined over time, and all the pastures had comparable beta diversity after treatment implementation. By this measure, beta diversity increased over time relative to the control in both patch-burn and modified rotational pastures compared to continuous grazing ($t_{PBG1}$ = 2.71, $t_{PBG2}$ = 3.45, $t_{Rotational}$ = 3.72, $p$ < 0.01 for all comparisons; Figure 5).

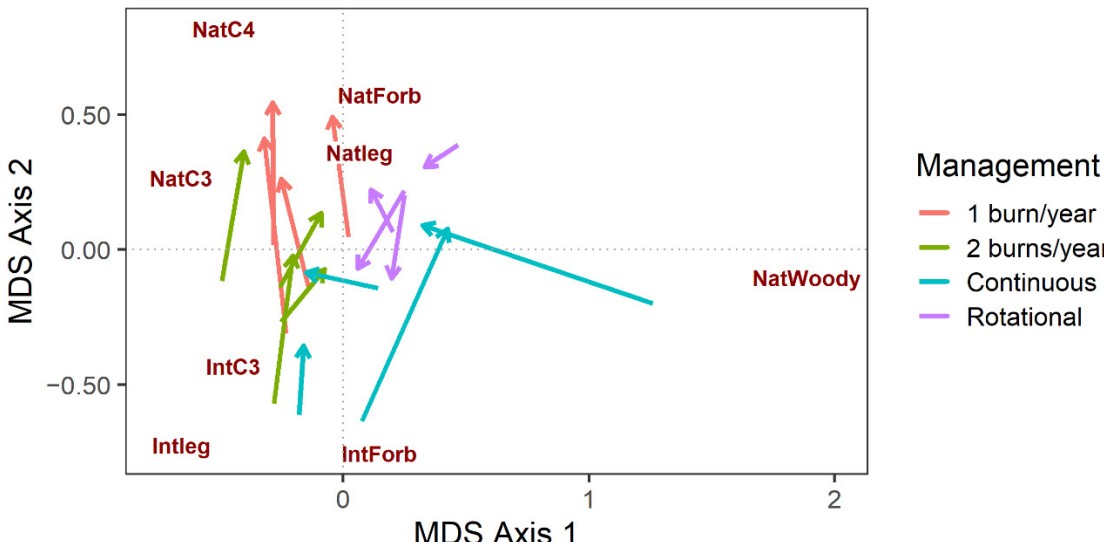

**Figure 4.** Nonmetric dimensional scaling ordination of vegetation functional groups in patch-burn treatments with one burn/year, two burns/year, continuous grazing, and modified rotational grazing systems at the Central Grasslands Research Extension Center in Streeter, ND. Arrows represent the trajectory of each pasture centroid from the beginning of treatment implementation until treatment completion, with arrow length indicating the magnitude of change. Nat = Native, Int = Introduced, Leg = Legume.

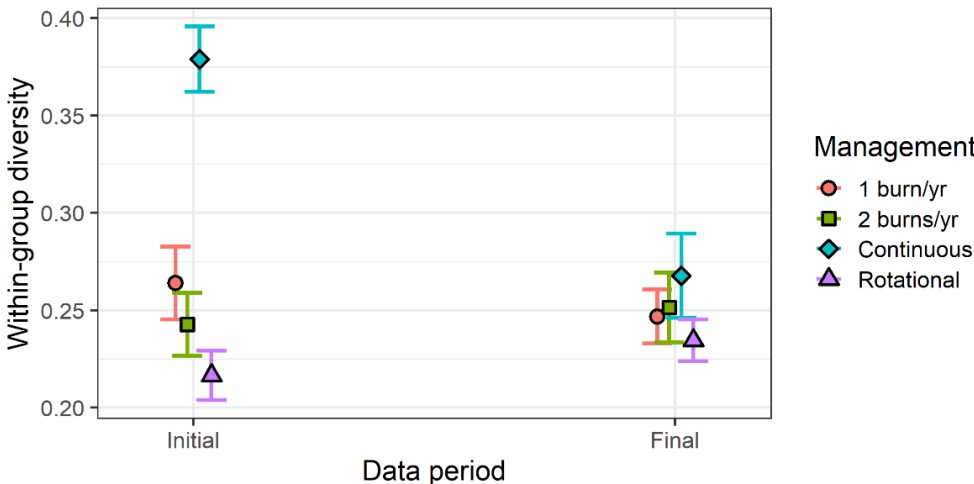

**Figure 5.** Changes in beta diversity (group area in ordination space) over time under four grazing regimes at the Central Grasslands Research Extension Center in Streeter, ND. Error bars represent one standard error.

Bare ground increased over the treatment duration relative to continuous pastures in PBG1 pastures (t = 2.91, *p* = 0.01; Figure 6). Litter cover decreased over the course of the study in PBG2 and modified rotational treatments relative to continuous pastures ($t_{PBG2}$ = −2.00, *p* = 0.07; $t_{Rotational}$ = −2.64, *p* = 0.02; Figure 6). Litter depth increases also occurred in all the treatments, but increases were not different relative to the continuous pastures ($F_{3,12}$ = 0.06, Figure 6). Robel pole values did not change significantly over the study period ($F_{3,12}$ = 0.51; Figure 6).

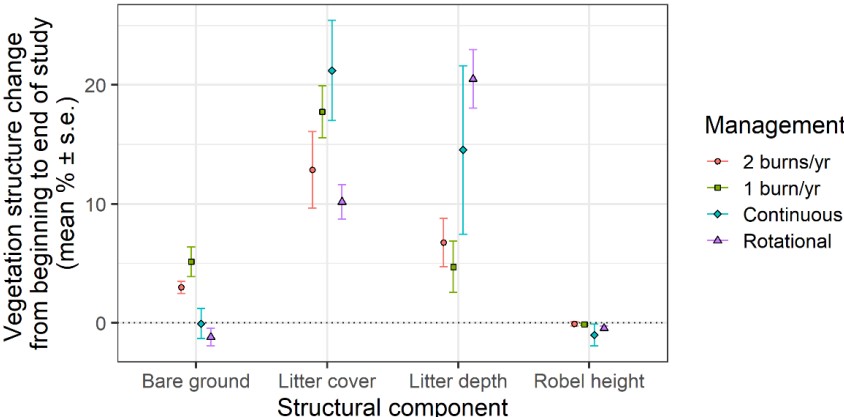

**Figure 6.** Changes in vegetation structure over time under four grazing regimes at the Central Grasslands Research Extension Center in Streeter, ND. Error bars represent one standard error.

Spatial heterogeneity was greatest in the PBG1 treatment compared to the other three treatments, and remained high throughout the study period (Figure 7). Litter cover spatial heterogeneity was again highest in PBG1 pastures at the beginning of the study period, but values were similar between PBG1 and continuous pastures by the end of the study (Figure 7). Heterogeneity in Robel height and litter depth tended to be low in PBG2 and continuous pastures (Figure 7). Modified rotational pastures had low spatial heterogeneity at the beginning of the study period, but heterogeneity increased over time, rivaling heterogeneity levels in PBG1 pastures by the completion of the study (Figure 7). Changes in spatial heterogeneity over time were not different relative to continuous grazing for bare ground and litter cover. However, visual obstruction and litter depth spatial heterogeneity increased in the rotational treatment relative to continuous pastures ($t_{\text{visual obstruction}} = 2.42$, $p = 0.03$; $t_{\text{litter depth}} = 2.59$, $p = 0.02$).

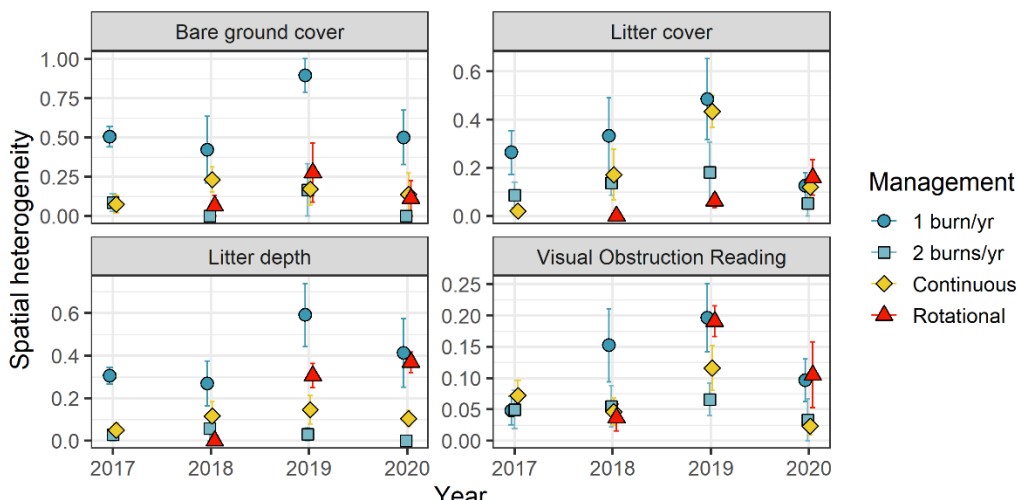

**Figure 7.** Spatial heterogeneity of vegetation structure over time under four grazing regimes at the Central Grasslands Research Extension Center in Streeter, ND. Error bars represent one standard error.

## 4. Discussion

Determining rangeland conservation management success in working landscapes requires simple, well-defined metrics [43]. While a logical goal for managers dealing with non-native species invasions may be to reduce or eliminate the cover of said species, the recalcitrance of many species to control measures [44] suggests it is more appropriate to evaluate alternative measures focused on enhancing ecosystem function and service

delivery. Following 3–4 years of alternative grazing management that included prescribed fire, we found little evidence of the lower cover of dominant non-native grass species. However, we did observe greater beta diversity in vegetation structure following patch-burn grazing and modified rotational grazing and greater spatial heterogeneity in modified rotational grazing compared to continuously-grazed pastures. Given the associations of vegetation diversity and structural heterogeneity with ecosystem service delivery and ecological function, managers should consider broader goals beyond simple cover decrease or eradication when facing non-native plant species invasions.

Areas heavily invaded by *Poa pratensis* and *Bromus inermis* typically have more continuous vegetation cover, a thick thatch layer, and deep litter [45,46]. Though dense vegetation structure is important for some grassland wildlife species (e.g., LeConte's sparrow, *Ammodramus leconteii* [47]), bare ground and sparse vegetation may be limiting in C3 grass-invaded rangelands [48]. Conventional management often seeks to reduce bare ground due to concerns of erosion and reduced forage [17], but more recent frameworks recognize the importance of sparsely vegetated areas for wildlife movement, foraging, burrowing, and basking, as well as critical habitat for imperiled plant species, such as *Penstemmon haydenii* [49,50]. PBG1 increased the area of bare ground compared to continuous grazing systems, increasing the availability of a limited resource for wildlife. Modified rotational grazing and PBG2 also decreased litter cover over the study period, providing more areas of lower vegetation structure. Patch burning removes litter and sets back woody cover through burning and subsequent herbivory, while modified rotational grazing may reduce grazer selectivity through high stocking density, forcing cattle to graze away areas with high litter [51,52]. It is also important to note that in a patch-burn system with rotating disturbances (and theoretically in our modified rotational system), any particular area of bare ground should be transitory, preventing the degradation that occurs in long-term 'sacrifice areas' under conventional grazing strategies [53]. Though bare ground as a limiting resource is a common phenomenon in conventionally grazed rangelands, it is likely particularly severe in *Poa pratensis*-invaded sites. Further research is needed to fully assess the limiting effects of excess structure on *Poa*- and *Bromus*-invaded systems with respect to livestock and wildlife.

Though patch burning and modified rotational grazing did not appreciably reduce the cover of *Poa pratensis* or *Bromus inermis* compared to continuous grazing, we saw increases in native forb cover in pastures under PBG1 management. While these results may seem contradictory at first, it is important to consider the mechanisms by which these non-native C3 grasses suppress competitors. Both species—*Poa* especially—form a deep layer of litter and thatch, which decreases soil temperatures and alters hydrological functioning and native plant competitive dynamics [3,54]. Reducing litter enhances forb expression [55], even if *Poa pratensis* itself is not killed by fire or grazing. Forb increase makes more floral resources available for pollinators [33], in turn enhancing an important ecosystem function despite a lack of non-native grass control. Increased native forb cover may also represent additional floral resource benefits, as introduced species, such as alfalfa and yellow sweet-clover, are relatively unattractive to native pollinators [56,57]. The reason for declines in forb cover in modified rotational pastures is currently unclear.

It is important to note that our results constitute responses to heterogeneity-based management practices within short timescales. Future work should consider changes to the structure and function of grasslands following long-term management in invaded systems. For example, a patch-burn study in invaded tallgrass prairie was ineffective at generating structural heterogeneity upon initial treatment application, but was able to achieve process-based management goals following additional rounds of treatment and stocking rate adjustments [58]. Although in our study the modified rotational system had low initial levels of beta diversity and spatial heterogeneity, these values mostly increased over time, particularly in spatial heterogeneity of vegetation structure and litter depth, indicating that this treatment may not have reached full effectiveness during the three years of study. However, small increases in invasive C3 grasses and decreases in forb

cover in the modified rotational treatment during the study period may warrant caution. It is also possible that, by using cover class categories instead of absolute percentages, we failed to capture slight declines in *Poa pratensis* and *Bromus inermis* cover over time. Future studies should consider alternate measures of cover and structure at a variety of spatial scales to fully capture structural responses to grazing treatments over short and long timescales [59,60].

As many managers recognize the impracticality of eradicating alien species in heavily-invaded areas, a new focus has become enhancing the functioning of altered landscapes to mimic the desired properties of uninvaded landscapes [61]. Though we did not directly measure ecosystem service delivery, numerous studies have linked vegetation structural heterogeneity to the diversity of many taxa reviewed in [16], temporal stability in livestock forage resources, and improvements to livestock forage quality [62]. Concurrent studies in these experimental pastures additionally found benefits of patch-burn grazing to avian diversity and nest densities [34,47], livestock forage quality [63], and floral resource diversity and abundance [33] associated with changes in vegetation structure following the implementation of heterogeneity-based management practices. Even in situations when complete eradication of *Poa pratensis* and *Bromus inermis* may be undesirable or logistically infeasible, it appears to be possible to generate heterogeneity—and some of the associated ecological service delivery—through grazing management.

## 5. Conclusions

These results show that despite minimal changes in *Poa pratensis* and *Bromus inermis* following three to four years of vegetation management, patch-burn grazing and modified rotational grazing with variably stocked paddocks generated increases in beta diversity, increased bare ground, decreased litter cover, and increased heterogeneity compared to continuous grazing. Many studies have already recommended that heterogeneity in vegetation structure should be a paramount goal in range management due to its influence on biodiversity and ecosystem services; this work extends the paradigm to the practical management of invaded landscapes. While we do not suggest that invasive C3 grass eradication efforts are not worthwhile, we show that the restoration of ecosystem structure and functioning is possible without eradicating non-native species *per se*. With limited resources in the face of unprecedented rates of ecosystem change and increasing demands on natural systems, process-oriented control measures based on simple structural metrics show the potential to guide range management efforts towards the restoration of desired grassland function in invaded landscapes. Our results allow future work to consider alternative metrics to measure the success of invasive species management, as well as the direct correlations between structural heterogeneity and ecosystem service delivery in invaded systems.

**Author Contributions:** Conceptualization, T.H., D.A.M., R.L. and K.S.; methodology, T.H. and D.A.M.; data collection, C.D. and M.W.; analysis, D.A.M.; original draft preparation, C.D.; review and editing, all authors; visualization, D.A.M.; funding acquisition, T.H., R.L., D.A.M. and K.S. All authors have read and agreed to the published version of the manuscript.

**Funding:** This research was supported by the Central Grasslands Research Extension Center and the USDA National Institute of Food and Agriculture, Hatch project number ND02367.

**Institutional Review Board Statement:** Not applicable.

**Informed Consent Statement:** Not applicable.

**Data Availability Statement:** Code and data is available at the data presented in this study are openly available in github at https://github.com/devanmcg/VegStructureComp, accessed on 1 February 2022.

**Acknowledgments:** We express our thanks to Megan Dornbusch, Brooke Karasch, Alex Rischette, Jonathan Spiess, and countless technicians for assistance in data collection.

**Conflicts of Interest:** The authors have no conflicts of interest to declare.

## Appendix A

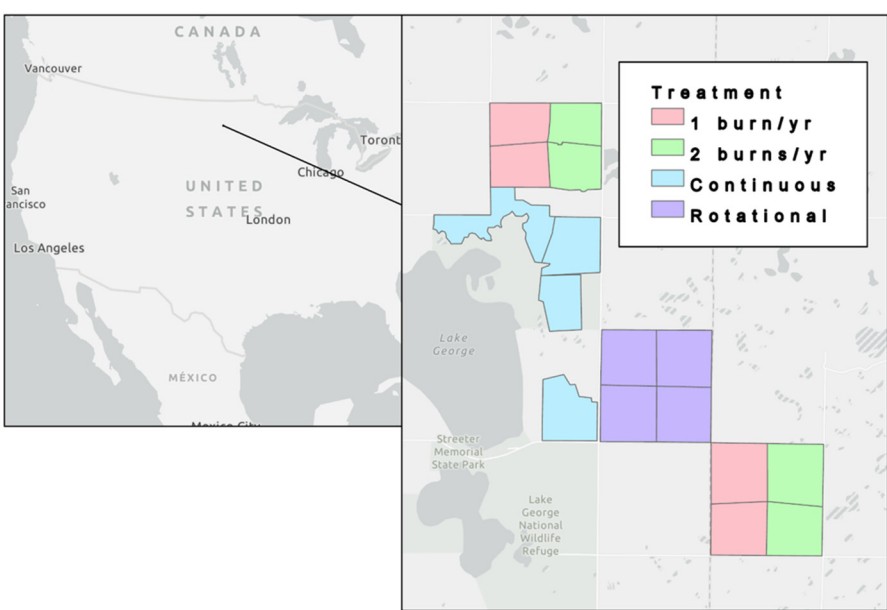

**Figure A1.** Location of study within the United States; (inset map) layout of experimental treatment pastures.

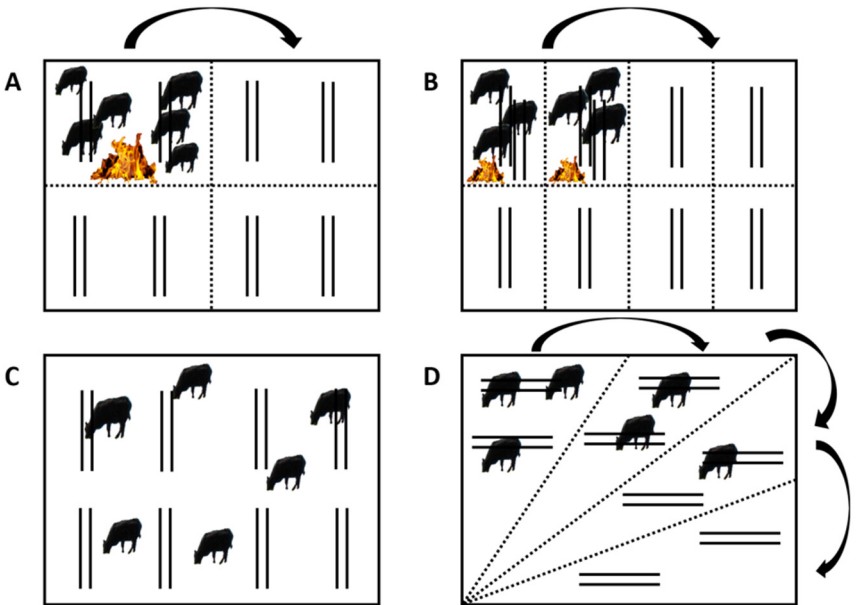

**Figure A2.** Depiction of (**A**) PBG1; (**B**) PBG2; (**C**) continuous; and (**D**) modified rotational treatments. Dotted lines indicate burn patch boundaries in PBG1 and PBG2 treatments, and the location of interior fencing in modified rotational treatments. Solid interior lines denote vegetation sampling transects. Arrows denote the rotation of burning from year to year in PBG1 and PBG2 treatments, and the rotation of stocking density in modified rotational treatments.

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
