# Peer review of "Heterogeneity-Based Management Restores Diversity and Alters Vegetation Structure without Decreasing Invasive Grasses in Working Mixed-Grass Prairie"

_land, doi:10.3390/land11081135_

Round 1

Reviewer 1 Report

Line 129, how the four grazing treatments were designed? I prefer a table that explains how they differ in grazing schema or burning schema. The head of the table may include treatment name, burning frequency, grazing intensity and other necessary information that make each treatment unique, or conversely, the table head can list the four treatments and table rows list the items that differentiate the treatments. Similarly, the data collect (2.2) would be included in the table so that data collection for each treatment could be easily seen.

How the replicates were decided for each treatment? were there any field difference (width and length, soil, vegetation, landforms, etc) among the replicates? How those 4*4 plots locate? A map would be helpful to explain those.

Lines 130-132, the abbreviations for PBG( PBG1, PBG2) should be given when they appear for the first time

Line 189-195: I think beta diversity, and spatial heterogeneity were not explained clearly, how they were computed? The definitions in lines 193 and 194 for  beta diversity, and spatial heterogeneity look very unclear.

Line 201-205, since Kentucky bluegrass means Poa pratensis and smooth brome is Bromus inermis, where the texts and Figure 1 use different names for the same thing? this would make readers confused. Texts and figures should be consistent

 Line 211 In figure 2, "Changes to the cover of smooth brome" but what here "changes" mean? The label for y-axis changes over study is not clear either. What "change"? Figures should be self explainable. Similarly labels  in many other figures (e.g., Fig 3) are not clear. I suggest authors go through all the figures and make sure they are clear even without looking for the explanations from the texts.

In the discussion section, I think it would be good to point out limitations in the definition and computation of the indices (e.g., uncertainties due to background environment or data collection). 

Reviewer 2 Report

The current study entitled „Heterogeneity-based management restores diversity and alter vegetation structure without decreasing invasive grasses in working mixed-grass prairie” is an interesting publication of a good professional standard, generating many questions for further work. Authors evaluate patch-burn grazing with one and two fires per year and variably-stocked rotational grazing in Kentucky bluegrass (Poa pratensis) and smooth brome (Bromus inermis) invaded grasslands using traditional (cover) and process-based (diversity and vegetation structural heterogeneity) frameworks in central North Dakota, USA. They sought to determine if we could mimic the heterogeneity-creating effects of fire and grazing without the use of fire via a modified rotational grazing system with variably-stocked paddocks. The experiment carried out for the study was well-organised.

In the title, alter should be in 3rd person singular.

Meaning of abbreviations used in this article (e.g. PBG, AUM, VOR) should be given at the first occurrence. Two treatments employed patch-burn grazing (PBG1 and PBG2)…

Abstract: The structure of this part of the manuscript should be improved to better highlight the scientific added value of the project. Based on the Abstract, one should be able to formulate a new hypothesis. The first part explaining the reasonability of the project can be shortened. The objectives can follow a single nice expanded sentence clarifying the scientific challenge that established the research plan. In the mid part, I propose to add numerical data, results of calculations. Then findings should be scientifically explained, and finally, scientific utilization should be added. Terms such as greater (appearing three times within three rows) and may favour do not support this.

Introduction: Knowledge gap is clarified. The objective of the study is obvious based on the introduction. Literature is up-to-date. Well structured, good professional quality. The objectives are correctly stated.

Materials and methods: Basically detailed, well structured. I recommend that it be supplemented with more detailed meteorological and soil data for the experimental areas.

Results: This part of the manuscript is well-structured and well-detailed, properly introducing the findings. Letter size of axes x and y could be close to that of the text to look nicer. In Fig. 7, connecting data points per years is meaningless.

Discussion: Findings are properly discussed and causal relationships are clearly described.

Conclusions: This section should be understandable alone, please, focus on your own findings and their explanations, reflecting each aspect as per the chapters of Results. Here further references should not appear, this part is the collection of the authors’ own conclusions based on their own results. This should be a reflection to the objectives one by one, as listed at the end of the Introduction. Furthermore, the added-value of the project should be given, referring to the knowledge gap covered by the study. In addition, future research options should be given. Based on this part of the article, one should be able to determine a new scientific problem and propose a new research plan.

Round 2

Reviewer 1 Report

I thank the authors for the revision and I think the paper is now good to publish.